# Evaluating Iodine-125 DNA Damage Benchmarks of Monte Carlo DNA Damage Models

**DOI:** 10.3390/cancers14030463

**Published:** 2022-01-18

**Authors:** Shannon J. Thompson, Aoife Rooney, Kevin M. Prise, Stephen J. McMahon

**Affiliations:** Patrick G Johnston Centre for Cancer Research, Queen’s University Belfast, Belfast BT9 7AE, UK; sthompson76@qub.ac.uk (S.J.T.); arooney25@qub.ac.uk (A.R.); k.prise@qub.ac.uk (K.M.P.)

**Keywords:** Monte Carlo, DNA damage, Monte Carlo damage models, Geant4-DNA

## Abstract

**Simple Summary:**

Simulation of initial radiation-induced DNA damage remains a major area of research, with numerous Monte Carlo models having been developed to model radiation effects on the cellular scale. While many models have been reasonably fit to a range of biological endpoints, there remains a lack of robust benchmarking data. Here, we investigate the application of a dataset on strand breaking in a single DNA strand through incorporation of radioactive Iodine-125 to distinguish between different Monte Carlo nanoscale physics models. We find that, while all models are able to effectively fit this data, they do so using significantly different best-fitting parameters and make substantially different predictions for other endpoints. These observations suggest that most nanoscale models broadly agree on the distribution of energy and can be made to fit to single datasets, but robust, multi-endpoint analysis is required to fully optimize and validate these approaches.

**Abstract:**

A wide range of Monte Carlo models have been applied to predict yields of DNA damage based on nanoscale track structure calculations. While often similar on the macroscopic scale, these models frequently employ different assumptions which lead to significant differences in nanoscale dose deposition. However, the impact of these differences on key biological readouts remains unclear. A major challenge in this area is the lack of robust datasets which can be used to benchmark models, due to a lack of resolution at the base pair level required to deeply test nanoscale dose deposition. Studies investigating the distribution of strand breakage in short DNA strands following the decay of incorporated ^125^I offer one of the few benchmarks for model predictions on this scale. In this work, we have used TOPAS-nBio to evaluate the performance of three Geant4-DNA physics models at predicting the distribution and yield of strand breaks in this irradiation scenario. For each model, energy and OH radical distributions were simulated and used to generate predictions of strand breakage, varying energy thresholds for strand breakage and OH interaction rates to fit to the experimental data. All three models could fit well to the observed data, although the best-fitting strand break energy thresholds ranged from 29.5 to 32.5 eV, significantly higher than previous studies. However, despite well describing the resulting DNA fragment distribution, these fit models differed significantly with other endpoints, such as the total yield of breaks, which varied by 70%. Limitations in the underlying data due to inherent normalisation mean it is not possible to distinguish clearly between the models in terms of total yield. This suggests that, while these physics models can effectively fit some biological data, they may not always generalise in the same way to other endpoints, requiring caution in their extrapolation to new systems and the use of multiple different data sources for robust model benchmarking.

## 1. Introduction

Radiation is well understood to induce cell death through damage to the DNA, enabling its use as a vital type of cancer treatment [1]. The mechanisms behind radiation interactions and subsequent damage have been extensively researched over the last century, supported by numerous studies characterising the yields and types of damage for a range of radiation types and biological targets [2]. There is now a good understanding of the qualitative dependence of radiation-induced damage on a range of factors, most notably particle type and energy, via their Linear Energy Transfer (LET) [3].

However, while it is widely accepted that radiation track structure plays a key role in DNA damage [3], robust quantitative predictions of radiation-induced damage remain elusive. A range of Monte Carlo track structure codes have been developed or extended to simulate radiation interactions on the cellular scale, including PARTRAC [4], RITRACKS [5], KURBUC [6] and Geant4-DNA [7,8]. These tools allow simulation of low-energy electron interactions, down to the ionisation threshold of water required when investigating the nanoscale interactions that influence DNA damage. By modelling the physics of radiation interactions, they can describe ‘direct’ DNA damage—that is, damage caused by energy deposition within the DNA structure itself. In addition, many of these codes have been expanded to incorporate subsequent radical chemistry in water, such as the production of OH, H_2_O_2_ and other reactive species. This enables the codes to also predict yields of ‘indirect’ DNA damage, defined as that which is caused by interaction of these reactive species with the DNA.

Many of these Monte Carlo codes have then been integrated with models of DNA to enable predictions of the yield of DNA damage by ionising radiation of different LETs, either as a primary endpoint or as part of more general radiosensitivity predictions [9,10,11,12,13,14,15,16]. In general, these models are in qualitative agreement regarding the overall trends in yields of radiation-induced DNA damage, although there remains significant quantitative differences between models. In addition, these models often contain a large number of free parameters which can be adjusted to fit to observed data. This includes key assumptions of the amount of energy which must be deposited in a DNA strand to cause a break, or the probability of an OH radical interaction leading to damage. The key features of these models have been reviewed elsewhere [17,18].

A significant challenge in this area is that different physical models show significant heterogeneity in the included physical and chemical interactions and their associated cross-sections. These differences can significantly influence the distribution of energy on the DNA scale, which can in turn modify the parameters required to reproduce a given biological effect. Together with the experimental limitations of most estimations of DNA damage, parameters often carry significant uncertainties, and in many models are selected with reference to those used in older studies, often using different Monte Carlo models and simulation geometries. With a lack of good-quality experimental data for individual strand breaking, benchmarking models remains a major challenge, particularly when it comes to the more complex damages which can be encountered in cells, such as multiple breaks in close proximity forming a clustered lesion.

One dataset which can be used to investigate the influence of different physical models is the work of Kandaiya et al. In this work, Iodine-125 (^125^I) was incorporated into a linear strand of DNA. Following its decay by electron capture, the distributions of strand breaks with distance from the decay site was evaluated [19]. Building on previous work by that group [20,21], this represents one of the few studies presenting a high-resolution distribution of damage around a known radiation source. This experimental work was previously modelled by Nikjoo, who showed that good agreement could be obtained with experimental results if it was assumed that the backbone of a base pair could be broken if the energy deposited within it by a decay was greater than 17.5 eV [22]. As this represented one of the first models reproducing such experimental data, this energy threshold is now commonly used across different models for the calculation of strand breaks [6,10]. However, in many cases, these models use different underlying physical Monte Carlo codes, and it is unclear if this threshold would translate effectively between different physical models. A better understanding of how this threshold must vary to fit the same experimental data and simulation geometry with other Monte Carlo models, such as Geant4-DNA [23], could provide an insight into the impact physical models have on the calculated DNA damage.

Within the Geant4-DNA toolkit there are various physical models for the simulation of low energy electron interactions in water. The three main physics constructors offered by Geant4-DNA are G4EmDNAPhysics_option2 (opt2), G4EmDNAPhysics_option4 (opt4) and G4EmDNAPhysics_option6 (opt6), whose main differences are summarised below and have been reviewed in more detail elsewhere [24], and in the Appendix A. In some applications, these differences in underlying assumptions lead to significant variations in energy depositions on the nanoscale, suggesting that optimising physical DNA damage models may require different energy and radical interaction thresholds (see Incerti et al. [24] and Table 1 within for further details).

To explore this, this work modelled the Kandaiya DNA damage data following the modelling approach used by Nikjoo, but employing Geant4-DNA, and in particular the three physics lists described above. Their relative performance was evaluated using both published damage parameters alongside the calculated best-fitting parameters. This enabled an exploration of the impact of different physical models on predicted radiobiological responses, and the relative importance of parameter tuning to model accuracy.

## 2. Materials and Methods

### 2.1. Experimental Data

This study made use of experimental data published by Kandaiya et al., investigating the distribution of single strand breaks (SSBs) following a decay of ^125^I incorporated into a strand of DNA [19]. Atoms of ^125^I were incorporated into the centre of a 41-base-pair molecule of double-stranded DNA with a known sequence, with the end of one strand labelled with an atom of ^32^P. These DNA molecules were incubated in either 20 mM PBS or 20 mM PBS + 2 M DMSO (low and high scavenging capacity, respectively) for up to 20 days, over which time approximately 20% of the ^125^I decayed.

After incubation, DNA samples were loaded into a high-resolution DNA sequencing gel and separated via electrophoresis. This process separated the fragments by size, with shorter, lighter fragments travelling further than heavier fragments, giving rise to characteristic bands on the gel. Autoradiographs of these gels were produced by imaging radioactive decays produced by the ^32^P bound to fragments to localize the regions of the gel corresponding to each fragment size. These sections were then excised from the gel, and the free ^32^P activity was quantified using a liquid scintillation counter to provide an accurate measure of the amount of ^32^P in each band, and thus the amount of DNA fragments of that size produced by the ^125^I decays. It is important to note that this measurement quantifies fragment sizes based on the distance from the ^32^P-labelled end of the DNA and the closest strand break. Any additional breaks occurring closer to the position of the ^125^I, for example from a clustered lesion, would not affect this fragment size and would not be detected.

These fragments were thus measured in terms of both the fraction of the total activity associated with fragments of each size (Figure 3 in Kandaiya et al. [19]) and through an attempt to predict the probability of break per decay based on this normalised data (Figure 5 in Kandaiya et al. [19]). Data from both these figures has been extracted for comparison with this work. It is important to note, however, that as only normalised fragment distributions are available, it is impossible to extract a unique probability distribution for breaks from a given fragment distribution, which will be discussed further below.

### 2.2. Simulation Setup and Scoring

Iodine decay and subsequent DNA damage were simulated in TOPAS 3.6.1 [25] with the TOPAS-nBio 1.0 [26] extension package to provide geometric structures. The simulation geometry was implemented following that described in previous studies by Nikjoo and Charlton [22,27]. In this approach, the DNA double helix is modelled as a central cylinder representing the bases, surrounded by a series of arches representing the sugar-phosphate backbone. The central cylinder has a diameter of 1 nm, while the outer bases have a total diameter of 2.3 nm. The backbones corresponding to each individual base have a thickness of 0.34 nm, and rotate around the central cylinder in 36° steps, giving ten bases per complete turn. This geometry is implemented in TOPAS-nBio as the “TsCharltonDNA” structure and is illustrated in Figure 1. Due to the design of the Geant4-DNA physics processes under consideration, all components of these DNA strands are modelled as water.

To simulate the energy deposition in this DNA strand for comparison to the experimental results of Kandaiya et al., a strand of 40 base pairs in length was created and embedded in a surrounding water box (20 × 20 × 40 nm^3^) to provide appropriate electron and free radical backscattering. For each event, a stationary atom of ^125^I was placed within the first base pair in the strand, centred within the base and offset 0.15 nm towards one of the backbone strands (designated as strand 1), as in previous studies [22,27]. This atom undergoes radioactive decay by electron capture to an excited state of ^125^Te, which further decays to the ground state. Both of these processes can lead to the emission of secondary electrons, through Auger emission for the ^125^I decay, and through internal conversation for the relaxation of ^125^Te. The interactions of these secondary electrons are then simulated within the DNA strand and surrounding water, and a tuple scorer was used to record the energy deposited within each backbone volume, along with relevant additional information such as the position of the energy deposition, the depositing particle and the origin of the depositing particle.

For these simulations, the standard TOPAS physics lists were modified to enable the investigation of the impact of different physics lists. Firstly, the “g4radioactivedecay” modular physics list was added to enable the simulation of the decay of ^125^I and ^125^Te. Secondly, the default EM physics list was replaced with either “g4em-dna_opt2”, “g4em-dna_opt4” or “g4em-dna_opt6”, corresponding to the respective Geant4-DNA physics constructors. Simulations were carried out for each of these three sets of constructors for all scenarios to enable a comparison between their predicted energy depositions and biological damages. Production cuts were set to 1 nm to ensure all secondary electrons were created.

In addition to these physics changes, Geant4-DNA’s chemistry simulation routines, as implemented in TOPAS-nBio, were also used to simulate the interaction of excited chemical species following the physics stage. This was accomplished by including the “TsEmDNAChemistry” process in the physics list, which made use of the default TopasChemistry diffusion and interaction parameters [28]. Radicals were generated in the water volume surrounding the DNA and allowed to diffuse freely until they either left the water volume or entered the DNA volume, or the maximum simulation time of 1 μs was reached. Those that entered the DNA volume (defined as having a radial position of less than 1.65 nm from the centre of the DNA strand) were immediately scavenged. Particles scavenged in this way were recorded in a tuple scorer, recording their chemical species, position, base pair and strand where the interaction occurred. These were then used to model free-radical induced damage, as described below.

For each of the three physics lists under consideration, 250,000 decays were simulated. Although the degree of variability within each event remains extremely large, with most bases seeing 0 energy deposit in the majority of decays, the final uncertainty in the average dose remains relatively small (<2% across the first 20 bases, <8% at most distant base simulated).

### 2.3. DNA Damage Models

DNA damage was modelled based on energy deposition and chemical interactions within the bases, on an event-by-event basis, taking a similar approach to that published by Charlton and Nikjoo [22,27]. In this approach, SSBs are split into two discrete classes—‘direct’, caused by energy deposition directly within a base, and ‘indirect’, caused by energy deposition within the surrounding water, leading to the generation of radicals, which can go on to interact with and damage the strand.

Direct damage is determined based on the amount of energy deposited within each backbone of each base following a single decay. The initial analysis considered strand one, which is closer to the incorporated ^125^I, as this is the strand quantified in the Kandaiya study. For each event, the energy depositions in each base were accumulated to give an array of the energy deposit per base per event. SSBs are then initially determined by a simple threshold model, where a break occurred if more than a certain amount of energy, $E_{thresh}$, was deposited, and lower amounts of energy deposit had no effect. This can be readily calculated for all simulated events, and then either the total probability distribution calculated by determining the rate of breaking across all events, or a fragment size distribution determined by obtaining the position of the break most distant from the ^125^I for each decay, as schematically illustrated in Figure 1.

As an initial value, $E_{thresh}=17.5$ eV was considered, as this is a value which has been widely used in a range of publications. However, in this analysis, this was found to agree poorly with the observed data. To obtain best-fitting parameters, $E_{thresh}$ was varied from 10 to 40 eV, and the goodness of fit to the observed data was quantified. Goodness of fit was determined based on the log-transformed fragment size distributions, using the sum-of-squares errors because no uncertainties are available on the Kandaiya data, that is: $SSE=\sum {\left(\log\left(f_{i}\right)-\log\left(F_{i}\right)\right)}^{2}$, where $f_{i}$ is the observed fraction of fragments of a given size and $F_{i}$ is the modelled fraction. The best-fitting value was then chosen as that which minimised this SSE  value. For all cases, break yields were only simulated for bases out to the most distant position measured by Kandaiya et al., as more distant, un-measured, breaks do not impact on the observed distributions, as described in more detail below.

In an alternative approach, rather than a discrete threshold, a probabilistic window method can be used. In this method, break probability is 0 below some lower threshold, and then increases linearly until it reaches 1 at some upper threshold. One example of such an approach, used in PARTRAC and some other applications, uses lower and upper thresholds of 5 and 37.5 eV [9,10]. The applicability of this approach was also investigated, using the same approach as above, as a function of window size (that is, difference between maximum and minimum energies), to determine how this impacted on predicted energy thresholds.

The yield of Double Strand Breaks (DSBs) can also be analysed for these systems. In this case, both strands were considered. Initial SSB calculations were performed for each strand, as described above, to give a list of bases subject to SSBs on each strand for each event. These two lists were then combined to determine if, at any point, both strands were broken within 10 base pairs of one another. If they were, this was then recorded as a DSB. This was used to investigate the impact of different distributions of breaks, even in cases where the absolute yield is similar.

The indirect effects of chemical species were investigated in a similar manner. In contrast to physical energy deposition, there were typically only a small number of radical interactions with the strand per decay. This focused on the OH radical, as this is believed to be the dominant contributor to indirect effects, and all other radical interactions were neglected [29]. OH radicals entering bases were accumulated per decay, giving an array of the number of radicals interacting with each base for each decay. Each OH interaction was then taken to have some fixed probability of causing a strand break, given as $P_{OH}$, independent of any physical interactions. This gave a probability of indirect damage for each base, based on the number of interacting OH radicals. $P_{OH}$ was optimised in a similar way, as described above, for the energy thresholds.

Data is available in the Kandaiya study for low and high scavenging conditions. To model this, in the high scavenging condition it was assumed that there was no indirect component to DNA damage. Thus, only direct damage by energy deposition within the strand was considered, and this was used to optimise $E_{thresh}$ for each model. In the low scavenging condition, both direct and indirect damage was considered—for this case, $E_{thresh}$ was taken from the fully-scavenged model to describe the direct physical damage component, and $P_{OH}$ was fitted as a single parameter. The results of these fits are discussed below.

### 2.4. Code Availability

The TOPAS parameter files used to simulate these results, together with a scoring extension used to customise the tracking and scavenging of OH radicals for this system, are available online at https://github.com/SJMcMahonLab/IodineDecayModel (accessed on 1 December 2021). These are also accompanied by a suite of Python tools which can be used to analyse the simulation output and generate data corresponding to the figures presented below.

## 3. Results

### 3.1. Energy Deposition and Radical Interactions

Each iodine decay produces a complex electron spectrum, with peaks between 20–40 keV representing inner-shell Auger events, 2–5 keV representing L-shell Auger events and a broad distribution <1 keV representing the complex distribution of outer-shell Auger events. On average, 25.9 electrons were produced per decay, with an average total energy of 19.9 keV released per decay. The total energy is in good agreement with that reported by Nikjoo (19.8 keV/decay), although the total number of electrons is higher than in their simulation (20.1/decay), suggesting some possible differences in low-energy electron yields. The average distribution of particles produced by iodine decay are unaffected by the chosen physics lists, although each individual decay may differ between each of the simulation runs.

Figure 2 presents a comparison of the energy deposited in different bases and the number of interacting OH radicals for each of the three physics lists under consideration. In terms of energy deposition, all three models show similar trends. Total energy deposited per decay varies significantly, ranging from 71 keV in opt2 to 106 keV in opt6, but this is dominated by the first few base pairs, with models converging at longer distances. From base pair 5 to the end of the strand, average energy deposited per decay between the three models agrees to within 5%.

A slightly more complex picture emerges in terms of OH radical interactions, with noticeably different trends among the models. In particular, opt6 begins with the highest rate of OH radical interaction at base 0 but falls to approximately half the rate of the other models by base 20.

It is important to note that these are average interaction rates, and in practice the majority of bases, particularly those distant from the site of decay, see zero energy deposition and radical interactions for most decays. Differences in the magnitude of energy deposit for the few cases where interactions do occur can lead to substantial impacts on the overall yields of damage, even when average energy deposits are similar, as will be seen below.

### 3.2. Direct SSB Yields

These models were benchmarked against predictions making use of the 17.5 eV threshold used in previous work, as well as through re-fitting to optimise predictions in these models. Predicted normalised yields of DNA fragments of different size (determined by the most distant break from the site of the iodine decay) were compared with the experimental results of Kandaiya et al. for fully scavenged data. In this comparison, it is assumed that in fully scavenged conditions there is no indirect component of damage via radicals, and all strand breaks are the result of physical energy deposition alone, enabling $E_{thresh}$ to be fitted as a single parameter. The results of these comparisons are shown in Figure 3.

It can be seen that, due to different assumptions and underlying physics models, these simulations do not agree with the Kandaiya et al. data when a 17.5 eV threshold is used. In particular, it can be seen that this significantly over-estimates the yield of damage, and thus fragments, at longer distances from the iodine for all three models. This also then translates into a reduction in the yield of damage in the vicinity of the iodine, due to more strands having multiple breaks.

By re-fitting this data, all three models can be brought into good agreement with the observed data. As noted in the figure legend, this requires a significant increase in threshold energy—to between 29.5 ± 0.25 eV for opt2, to 32.5 ± 0.25 eV for opt6. In all three models, agreement with the overall trend is good, with the exception of a slight over-estimation of the yield of fragments recorded as having breaks at the site of the iodine. This agreement is comparable to that reported in previous work for physical damage [22], despite the significantly higher energy threshold.

Moving from a single energy threshold to a range, as used in PARTRAC and other approaches, does not significantly change the observed trends. Best-fitting thresholds for different ranges are shown in the Appendix A, showing a small gradual increase in median energy with increasing range, but no significant difference in fit quality between the models.

### 3.3. Indirect SSB Yields

The results of incorporating indirect damage via OH interactions with the backbone are shown in Figure 4, compared with the results of Kandaiya et al. in a very low scavenging environment. Here, direct damage is calculated for each of the models using the fit thresholds shown in Figure 3, and indirect effects were fitted separately. Specifically, a single probability value, $P_{OH}$, was fitted for the probability that an OH radical interacting with a strand would lead to damage at that site.

There is reasonable agreement between the modelled indirect damage rates and direct effects. The significant increase in breaks for distant sites is clearly reproduced for all models, increasing from approximately $3\times 10^{-3}$/decay for direct damage alone to $1.5\times 10^{-2}$/decay for direct and indirect damage in combination. However, in all simulations, a gradual trend is visible from breaks 6 to 15, whereas the observed damage approximately plateaus. This may be the result of statistical issues in the data, simplifications made in the radical chemistry model, or potential longer-range or cross-fire radical interactions resulting from energy deposition events which would occur outside the current simulation volume. However, due to a lack of robust uncertainty information in the experimental data, it is difficult to evaluate this in more detail.

When considering the interaction rate, both opt2 and opt4 had an identical best-fitting rate of $P_{OH}=0.16\pm 0.01$, which is reasonable given the good agreement between their long-range radical yields in Figure 2. Notably, this interaction rate is the same as the effective rate in Nikjoo et al., despite the very large differences in physical interaction thresholds. By contrast, opt6 requires a much higher interaction rate than the other models, due to its reduced radical interaction rate at long ranges.

It is important to note that the obtained interaction rate depends strongly on the chosen energy threshold for damage. In general, reducing the threshold for damage also reduces the best-fitting interaction rate, as more direct SSBs require fewer indirect SSBs to obtain the correct level of damage in distant bases. For example, if a threshold of 17.5 eV is used for all models, $P_{OH}$ becomes 0.12, 0.05 and 0.16, respectively. However, in all cases, significantly changing the direct damage threshold from the best-fit also significantly worsens the fit, with best-fits for 17.5 eV with indirect damage still having SSE 3–4 times higher than fits optimized for both parameters.

### 3.4. Uncertainties and Fitting Limitations

These data together appear to suggest that all three models perform equally well at describing this system and can be brought into good agreement through modest adjustments in relevant fitting parameters. However, because the experimental approach of Kandaiya et al. does not provide an absolute fragment yield, but instead a normalised distribution of the relative probability of each fragment size, this serves to obscure a number of significant differences between models which cannot be resolved through simple parameter adjustment.

In particular, because these data are normalised, they are not sensitive to the total yield of damage. This gives rise to a relatively broad minimum in the relationship between the fit residuals and the SSB threshold, suggesting that, while the minimum is well-characterised, a broad range of values are at least similarly compatible with the observed fragment distributions. These curves are illustrated for the three models in Figure 5A. These minima occur at a broadly similar position, despite relating to significantly different yields of strand breaks for each model. The SSB/decay and DSB/decay trends are shown in Figure 5B, showing substantial differences in yield between the models at the same energy. Notably, at the thresholds which best fit the observed data, the yield of SSB/decay differs by nearly 70%, with yields of 0.712 ± 0.002, 0.895 ± 0.002 and 1.207 ± 0.002 SSB/decay for models opt2, opt4 and opt6, respectively. Significantly, even allowing for the flexibility afforded by the normalisation of total break yields in the fragment distribution, there is no single set of fitting thresholds which provides good agreement for both yield of SSBs and distribution of SSBs across the three models.

One way to help resolve this issue and potentially clarify which model offers the best description of the true behaviour of this system would be through examining the absolute yield of SSBs. This can be readily evaluated from the simulations, as the exact distribution of all breaks is known. However, in the experimental approach of Kandaiya et al., only the last fragment is detected. The authors attempted to control for this by correcting the yield of breaks at each site for the yield of breaks at all more distant sites in an iterative fashion. Unfortunately, because the normal data is under-specified and degenerate with respect to this operation, the absolute yields cannot be uniquely resolved. This is illustrated for the case of opt2 in Figure 6.

When corrected, the Kandaiya et al. data shows a dramatic increase in yield of breaks at sites closer to the location of the ^125^I, with a 100% probability of causing a break at the site of the iodine. By contrast, although the yield of SSBs predicted from physical interactions in opt2 also increases, it peaks at a significantly lower value, at approximately 0.27. This seems to represent a significant disagreement between the model and experimental data. However, if, instead of taking the absolute break probabilities generated from the simulations, the fragment size distribution is converted into a break probability through the approach of Kandaiya et al., it can be seen that this yields a distribution which now peaks at a 100% break probability, even though this does not reflect the true underlying distribution. Similar behaviours are seen for opt4 and opt6, and indeed this is true for any fragment distribution as a consequence of this normalisation approach (see Appendix A for mathematical details). As a result, it is not possible to use this data to predict or define absolute yields to provide better estimates of damage thresholds.

### 3.5. Interaction Energy Distributions

It was noted in Figure 5 that there were noticeable discontinuities in the break yield and fit residuals at particular energies. These differences between the physical models can be understood by considering the amount of energy deposited by different interactions between the emitted electrons and the DNA strands. These distributions are illustrated in Figure 7.

The left panels compare the energy deposited, per electron interaction, for each of the three physics models. All three models have some structural similarities—there are a number of distinct peaks at higher energies, corresponding to the distinct ionisation potentials of water, together with a broader distribution of lower energy interactions depositing less than 10 eV. There are quantitative differences between these models, however, including small differences in the energies assumed to be deposited by each type of event, and large differences in the absolute rates and distributions across the different models. Of particular note, the higher energy individual ionisations can be seen to relate to discontinuities in the break yield plots, such as the significant drop in SSB yield if the threshold energy goes above ~32 eV for each model in Figure 5.

The lowest energy bin (<0.25 eV) has not been plotted in these histograms. This is because there are very large numbers of interactions depositing very small amounts of energy for both opt2 (vibrational excitation) and opt6 (small amounts of energy deposited during ‘elastic’ scattering events). These represent the majority of total interactions (>60% and >90% of events, respectively), but less than 1% of the total energy deposit, and so do not impact on the yield of damage.

The right panels show the distribution of energy seen in the backbone 10 bases away from the site of the iodine decay, to illustrate how these events combine to cause damage. The structure relating to the individual high-energy ionisations can be clearly seen, together with a broader background relating to multiple lower-energy interactions. Despite the average energy deposit being similar, as shown in Figure 2, there are noticeable differences in both the total rate of interactions (with 5.0%, 4.6% and 3.6% of decays depositing a non-zero amount of energy in base 10 for opt2, opt4 and opt6, respectively) as well as the distribution of energy (with average deposits of 14.9, 18.5 and 21.0 eV for opt2, opt4 and opt6, respectively). These differences in energy distribution give rise to the differences in break yield and threshold, as discussed elsewhere, even for similar average energy deposition per decay.

The importance of the peak of events around 32 eV can be seen in the SSB fits above, where all three physics models produce energy thresholds close to this ionisation energy, as stepping above it leads to a significant drop in break yield. Similar factors may have arisen in previous work, with the 17.5 eV threshold being positioned close to the second-highest energy deposition in water.

## 4. Discussion

Improved modelling of radiation-induced DNA damage remains an ongoing challenge. A broad variety of approaches have been applied, but there remains significant heterogeneity in underlying model assumptions and generated predictions, which remains difficult to resolve. By comparing the predictions of three physics constructors in the widely-used Geant4-DNA Monte Carlo toolkit, this work has highlighted some of the remaining challenges in this area.

All of the applied Geant4-DNA models were able to produce predictions which agreed with the experimental fragment distribution data with a high degree of accuracy, with no clear indication of a better-performing model. However, the energy thresholds required to do so differed significantly. This is true within the Geant4-DNA models (a variation of approximately 10% from lowest to highest), but it is much more striking when compared to other approaches, requiring nearly 70% more energy to be deposited per break than the best fitting parameter of the original analysis by Nikjoo [22]. Interestingly, this pattern was slightly different for indirect effects, with two of the models agreeing very well with the published parameter for OH-induced base damage, with only opt6 differing, again by approximately 70%.

In many ways, this observation is unsurprising, given that there are quite significant differences between the underlying model structures, their assumptions, and their input parameters. However, it is useful to examine this in a quantitative fashion because many published DNA damage models have applied parameter sets from other work without taking into account the potential mismatch between their physical damage code and these parameter values.

A detailed consideration of the fitting to this data also serves to highlight some of the challenges in robustly determining best-fitting values for these biological parameters. Perhaps the most significant of these remains the challenge in linking from experimental data—expressed in terms such as DNA fragment production or foci formation—to yields of double strand breaks in absolute terms. Even in an apparently well-defined system such as this Iodine decay model, inherent normalisation and the lack of an absolute reference obscure the relationships with the true quantities of interest. Similar issues are present across the literature, with extremely large variations in estimated DSB yields for different radiation types, based on the method used to quantify [30].

As a result of this, while the best-fitting threshold can be readily determined for this particular dataset with a high degree of accuracy, Figure 5 highlights that the minima of the fit is relatively broad, with a wide range of thresholds providing at least reasonable agreement with the fragment size distribution. This is despite these different thresholds giving rise to very different behaviours in other metrics, such as the total yield of DSBs. Similar issues are present in many other datasets, where the selection of data and fitting parameters gives significant flexibility in model development. This suggests that significant care must be taken in understanding the uncertainties associated with different experimental endpoints, and ideally fitting to a range of different types of data to provide more robust tests of fit predictions—for example, features of break complexity and distribution in addition to yield.

Consideration of the distribution of energy deposits in Figure 7 also highlights some potential challenges in fitting energy depositions to observed damage data. While the models deposit almost identical amounts of energy within the strands after the first few base pairs (Figure 2), how this is distributed differs quite significantly, with some models showing much higher energy deposition, and others showing more frequent smaller deposits. Notably, a large fraction of volumes which experience a break are associated with the higher ionisation energies of water (around 17 and 32 eV in these models). However, these ionisation potentials are characteristic of water, while DNA is actually made of a range of other biological materials, including the bases, backbone strands and a range of other associated biomolecules. Each of these components has an entirely different chemical structure, which means that while water may reasonably approximate the total rate of energy deposition, it will not effectively reproduce these new ionization potentials and interaction distributions. As a result, it is important to recall that any empirically-derived thresholds are likely correlative, at best, with the true relationship between energy deposition and radiation-induced damage in DNA. To address this, DNA damage models incorporating more realistic materials are required. Early work in the simulation of interactions in the composition of individual bases already shows clear differences in the many physical interaction parameters which may impact on DNA damage [31].

A further consideration around the relationship between energy depositions and DNA damage is the irradiation geometry used. In this work, we have used TOPAS-nBio’s recreation of the original Charlton DNA geometry to enable comparison with previous models of this system [22,27]. However, DNA geometries have been applied in other approaches. In many cases, these involve both different physical shapes to the regions designated as ‘backbones’ and different associated volumes. This can impact significantly on the energy deposited in backbone volumes for a given irradiation, and thus on best-fitting damage thresholds [32]. This may explain differences in best-fitting break thresholds reported in other works which use more complex models which attempt to more accurately mimic the true structure of DNA, but which reduce the volume of some of these components [4,10].

## 5. Conclusions

This analysis shows that a range of different electron physics models implemented in Geant4-DNA are able to effectively replicate observed data on the distribution of strand breaks following incorporated ^125^I decay, if SSB energy thresholds and OH radical interaction probabilities are adjusted accordingly. More detailed investigation of these parameters indicates significant physical and biological uncertainty, with strong dependencies on features which are likely artefacts of model assumptions rather than reflecting true biology. Addressing this, together with higher-quality experimental data on multiple DNA damage endpoints, is essential to refining these models and mechanistically validating them.

## Figures and Tables

**Figure 1 cancers-14-00463-f001:**
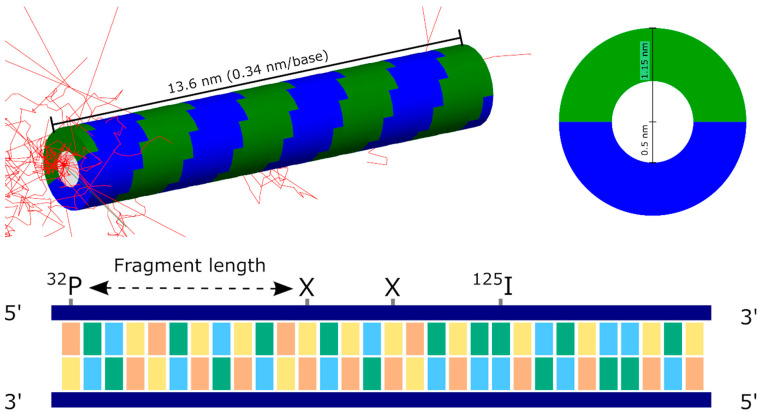
Illustration of simulated DNA geometry. (**Top left**): Visualisation of simulated geometry in TOPAS-nBio. A 40-base-pair fibre is generated, consisting of bases (white) surrounded by two semi-circular sugar-phosphate backbone strands (coloured green and blue). An iodine decay at one end generates a large number of secondary electrons (red) which deposit energy in the strand. This deposition is greatest in the immediate vicinity of the iodine, although some energy is deposited throughout the strand. (**Top right**): Cross-section of DNA fibre showing base and backbone radii. (**Bottom**): Schematic illustration of DNA fragment and fragment size determination. The assay of Kandaiya et al. detected the fragment size attached to an incorporated atom of ^32^P placed at one end of the strand. Following the decay of ^125^I, strand breaks are created (marked by X). The observed fragment size is determined by the distance from the ^32^P to the closest strand break, meaning the assay is insensitive to any further breaks created closer to the ^125^I.

**Figure 2 cancers-14-00463-f002:**
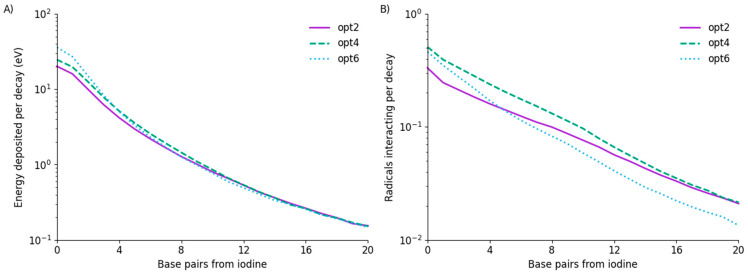
Physical energy deposition and radical interaction as a function of base. (**A**) Energy deposited in each base on one strand per decay. Base 0 contains the iodine. Average energy deposited differs between the models by approximately 50% at base 0 but rapidly comes into agreement after the first few bases. OH radical interactions with strands (**B**) are also of similar magnitude but show differing patterns for each model, which do not simply follow the physical trends.

**Figure 3 cancers-14-00463-f003:**
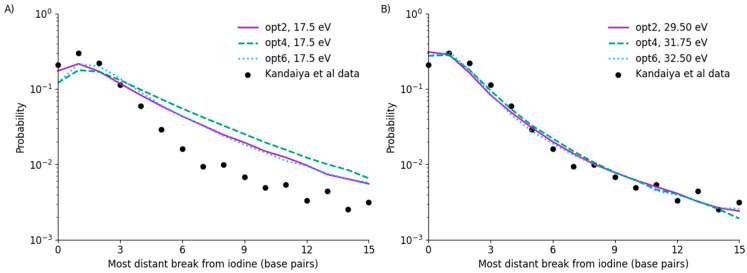
Predicted yields of DNA fragment sizes based on energy deposition alone, compared to the data of Kandaiya et al. for fully scavenged conditions. It can be seen that using a 17.5 eV threshold significantly over-estimates the yield of SSB and thus fragments starting at longer distances from the decay (**A**). Re-fitting this data yields good agreement with the experimental data for all models (**B**), although this requires a significant increase in the threshold energy (denoted in legend).

**Figure 4 cancers-14-00463-f004:**
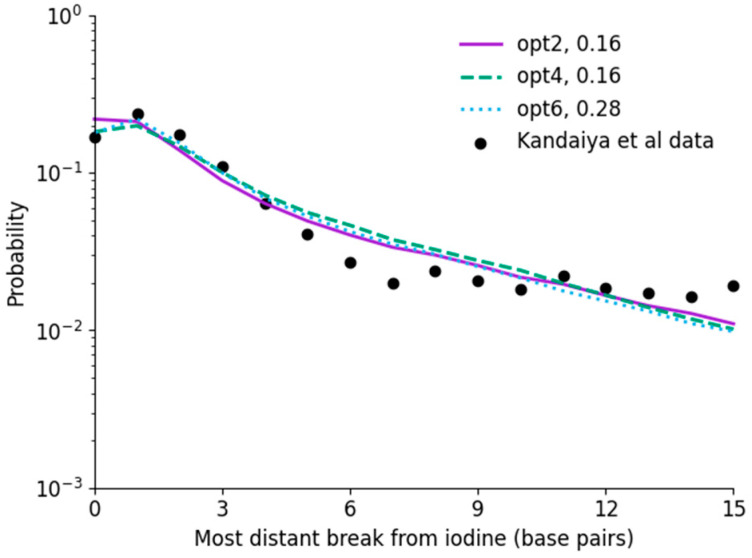
Analysis of chemistry fitting to Kandaiya et al. data for unscavenged solutions, fitting $P_{OH}$ to describe indirect effects, with best fitting values indicated in the legend. Good agreement is obtained, with interaction rates for opt2 and opt4 identical to that fitted by Nikjoo et al., while opt6 is significantly higher. While models all agree, there is a difference in trend between model and experiment at long ranges, with a continuing fall in the simulated data compared to a plateau in the experimental data. This may result from uncertainties in the experimental data, or longer-range interactions not currently encompassed in the simulations.

**Figure 5 cancers-14-00463-f005:**
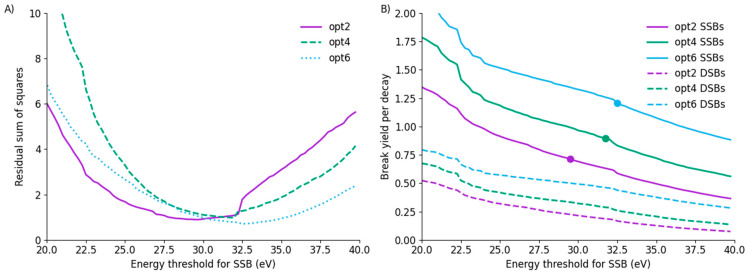
(**A**) Residual sum of squares as a function of SSB energy threshold for all three models. All three models have minima around 30 eV, although there is a broad range of similar-performing values due to the normalisation of data with respect to total break yield. (**B**) Yield of SSB and DSB as a function of energy for all three models, with SSB yield at best-fitting thresholds marked by a point. Despite average energy depositions per decay being similar across the models, there is a substantial difference in total break yields, for both SSB on the strand containing the iodine as well as DSBs. Significant structure can be seen in these curves, which reflects the distribution of energy deposited by different events in the different models.

**Figure 6 cancers-14-00463-f006:**
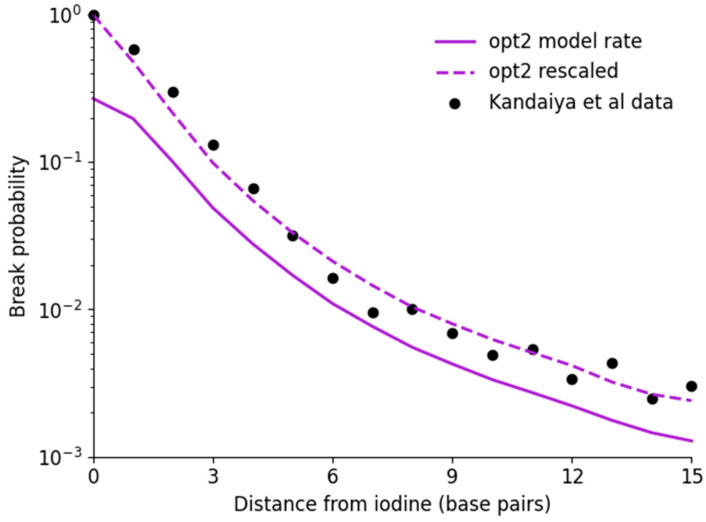
Impact of break probability scaling on predicted yield of SSBs. In Kandaiya et al., a normalisation technique was used to convert from a fragment distribution to an absolute probability distribution (points). This yield is significantly greater than that predicted by the model for the threshold energy which maximizes agreement for the fragment size distribution, which can be extracted directly. However, if the break probability is instead extracted from the modelled fragment size distribution using the same technique in Kandaiya et al., much better agreement is obtained, highlighting the ambiguity in these estimated total rates.

**Figure 7 cancers-14-00463-f007:**
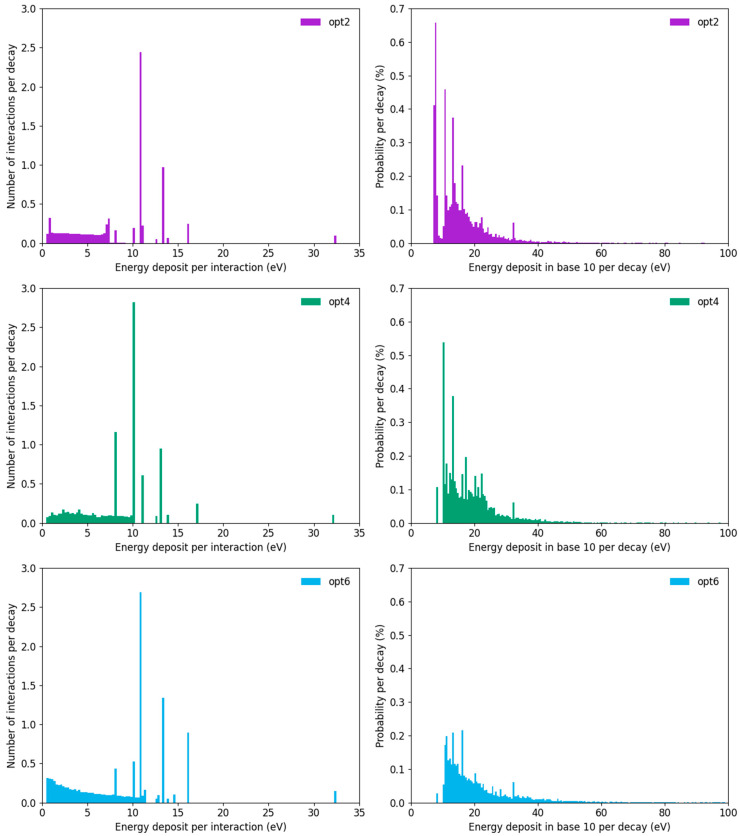
Histograms of energy deposition events from different models. (**Left**) Distributions of energy deposited per interaction are shown for each of the three models (opt2–opt6, top to bottom). In all cases, models have a continuous low-energy background (up to 7–12 eV, depending on the model) together with distinct peaks relating to particular ionisation energies in water. (**Right**) Amount of energy deposited per decay 10 base pairs from iodine. Here, substantial differences can be seen in both the number and distribution of events, with opt2 having significantly more, though lower-energy, events than opt6, even though average energy deposit varies by less than 10% between the models. Note that the vast majority (>95%) of events in all three models deposit negligible energy in this base and are not plotted for clarity.

## Data Availability

Full details of the model code used to generate the data in this paper, together with relevant analysis tools, is available online at https://github.com/SJMcMahonLab/IodineDecayModel (accessed on 1 December 2021).

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
