# Peer review of "Evaluating Iodine-125 DNA Damage Benchmarks of Monte Carlo DNA Damage Models"

_cancers, 2022, doi:10.3390/cancers14030463_

Round 1

Reviewer 1 Report

Manuscript Summary:

This present work study the DNA damage distribution following the decay of an incorporated 125I included in a short DNA strand by Monte Carlo simulation using three different physic models available in Geant4-DNA. Direct and indirect DNA damages were calculated and compared to the experimental data of Kandaiya and al. This work highlights that even if all the models can fit the experimental data, different best-fitting parameters have to be used. The results obtained in this paper also suggest that these parameters would be different if other biological endpoints were used. The paper highlight also the difficulty to compare/validate simulated to experimental data due to a lack of robust benchmarking data remaining a major challenge of this area of research.

Comments:

Introduction:

This introduction highlights the objectives of this paper but is a bit long. As the physical models developed in Geant4-dna are already well defined in the literature maybe the paragraph describing the differences between these three models could be moved in supplemental data.

Material and methods:

Figure 1: The dimensions of the different elements of the DNA fiber should be added on figure 1 (top) as well as the box containing the fiber piece (with its dimensions).

For the geometry, you have modelized 40 base pairs but in our results, you only considered the first 20 base pairs. This choice should be explained more clearly in the manuscript.

It is mentioned that the uncertainty in the dose is less than 2% across the first 20 bases. What is the uncertainty after these first 20 bases? I suppose that this uncertainty increases as one moves away from the decay site, is this the main reason why your results are focused on the first part of the fiber?

Results and discussion:

Energy deposited and radical interaction:

  • At the end of the first paragraph of the section you mentioned that the physics lists used don’t affect the number of particles produced, did you find exactly the same number of particles or slight variations between the different physics list?
  • In figure 2, the energy deposited by decay in each base concern only the physical stage or the physical and chemical stages?
  • For chemical interactions you only considered OH radicals, you show different trends among the models. Would these trends change if other radicals were considered? Have you ever made calculations taking into account other types of radicals?

Direct SSB yields: you show that the energy threshold must be increased (17.5 keV to 29,5, 31,75 or 32,5 keV depending on the physics lists used) to fit the Kandaiya data for fully scavenged conditions. So at this step you make the assumption that the specific experimental conditions are close to direct damages (physical stage). This point should be discussed.

Indirect DSB yields: if I understand correctly, in this section the direct damage has been calculated using the energy threshold defined in the section "direct SSB yields" i.e. 29,5, 31,75 or 32,5 depending on the physic list used. Then different OH probabilities were determined to fit the Kandaiya data. Have you evaluated the OH probability using an energy threshold of 17.5 keV? what is the impact of using different thresholds for the physical stage? This point should be discussed in the manuscript.

All simulations were carried out in water, so all the free parameters (energy threshold, OH probability…) were determined for this material depending on the physic list used. In the literature, the importance of taking into account the composition DNA base pairs for this type of simulation has been highlighted. Indeed, for example, the paper of S. Zein et al (NIMB 2021) show the differences with water simulation and emphasize the importance to take into account the DNA base pairs composition to better estimate the radiobiological damages. I think this point should be mentioned and discussed on this manuscript.

You mentioned also the importance of the geometry used for simulations. It would be interesting to perform the same study with different geometries in order to quantify the impact of the geometry used on the number of damages.

Author Response

We would like to thank the reviewer for their positive feedback on the manuscript, and their useful suggestions. We have revised the manuscript with respect to these suggestions, as described below. Changes in the updated manuscript are highlighted in red for clarity.

Introduction:

This introduction highlights the objectives of this paper but is a bit long. As the physical models developed in Geant4-dna are already well defined in the literature maybe the paragraph describing the differences between these three models could be moved in supplemental data.

As recommended, text on page 3 describing specific model features has been moved to supplementary information, and remaining text lightly edited for flow.

Material and methods:

Figure 1: The dimensions of the different elements of the DNA fiber should be added on figure 1 (top) as well as the box containing the fiber piece (with its dimensions).

Figure 1 has been updated with dimensions and a new cross-section illustration to show these more clearly. The bounding box is so much larger than the fibre that it’s not possible to include in the image without making the fibre too small to see clearly, so it has been omitted.

For the geometry, you have modelized 40 base pairs but in our results, you only considered the first 20 base pairs. This choice should be explained more clearly in the manuscript.

When the simulation was originally designed, we had not fully explored the impact of the normalisation inherent in the experimental technique of Kandaiya. We had originally thought that more distant damage may have a small, indirect, impact on observations, but when fully analysed it turned out that this has no impact at all on any observed quantity, so we did no further analysis on these distant breaks (the overall pattern is also largely unremarkable, following the same slow declining trend with distance).

A brief comment on this point has been added at the top of page 6 for clarity.

It is mentioned that the uncertainty in the dose is less than 2% across the first 20 bases. What is the uncertainty after these first 20 bases? I suppose that this uncertainty increases as one moves away from the decay site, is this the main reason why your results are focused on the first part of the fiber?

It's correct that the uncertainty increases with increasing distance from the site of decay, with the variance scaling approximately in proportion to the average energy deposit in the base, as expected from Monte Carlo statistics. The relative dose falloff is more gradual from base 20 to 40 compared to the inner bases, so the impact at this range is not so significant – with maximum uncertainty of 7.6% in the mean dose in base 40 across all models. This has been noted on page 5 alongside the other uncertainty measurement.

Results and discussion:

Energy deposited and radical interaction:

At the end of the first paragraph of the section you mentioned that the physics lists used don’t affect the number of particles produced, did you find exactly the same number of particles or slight variations between the different physics list?

The radioactive decay code in Geant4 is entirely separate to the EM interaction physics code, so there is no direct interaction between the two processes, and if only the decay and no subsequent interactions are simulated, exactly identical particle spectra are emitted regardless of the physics list.

However, because of the nature of the random number generator in Geant4, once interactions have occurred the RNG will be in different states for each subsequent decay for different physics lists, so the actual decay events will be different in each simulation, although the average long-run behaviour is still expected to be identical. This sentence on page 7 has been edited to clarify this point.

In figure 2, the energy deposited by decay in each base concern only the physical stage or the physical and chemical stages?

This is determined from the physical stage only. Due to matching the method used in previous work for tracking radicals (scavenged as they enter the volume) the energy tracked in the chemical stage is 0. This follows the conceptual approach in prior work, where the possible impact of chemical interactions (including any energy deposit) is covered by the chemical interaction term, rather than integrating into physical energy deposit term.

“Physical” has been added to the Figure 2 legend for clarity.

For chemical interactions you only considered OH radicals, you show different trends among the models. Would these trends change if other radicals were considered? Have you ever made calculations taking into account other types of radicals?

As noted in the methods (page 6), we focused on the OH radicals as there is good evidence that these are the major driver of indirect strand breaking, and that other radicals have a much smaller interaction cross-section. Similar approaches are taken by most other work, and so we felt this was a suitable comparison.

While we did not directly score other radicals in this work, from other similar analyses the density of most radicals tends to be initially similar for low-LET irradiation as considered here, so it is unlikely they would lead to significant changes in break distributions. However, this cannot be meaningfully evaluated in this data as radical-specific interaction rates would not be possible to fit to a dataset with only a single scavenging condition. Multiple conditions, ideally with multiple scavengers, might enable this to be resolved, but to our knowledge no such data is available. 

Direct SSB yields: you show that the energy threshold must be increased (17.5 keV to 29,5, 31,75 or 32,5 keV depending on the physics lists used) to fit the Kandaiya data for fully scavenged conditions. So at this step you make the assumption that the specific experimental conditions are close to direct damages (physical stage). This point should be discussed.

That’s correct, for full scavenging we assume as in previous work that the indirect component of damage is negligible. This was briefly mentioned on page 6 at the end of the methods, but based on feedback this section has been expanded and clarified slightly, and text at the end of page 7 & beginning page 8 updated to re-emphasise which data is being considered and the associated assumptions.

Indirect DSB yields: if I understand correctly, in this section the direct damage has been calculated using the energy threshold defined in the section "direct SSB yields" i.e. 29,5, 31,75 or 32,5 depending on the physic list used. Then different OH probabilities were determined to fit the Kandaiya data. Have you evaluated the OH probability using an energy threshold of 17.5 keV? what is the impact of using different thresholds for the physical stage? This point should be discussed in the manuscript.

Thanks for this interesting suggestion, this is something we hadn’t investigated in detail previously. We re-ran the fits for the radical damage rate using the original threshold of 17.5 eV to evaluate the impact. We saw that the OH radical interaction rate significantly decreased, likely reflecting the lower numbers of SSB needed to occur at more distant positions to give the correct trend. The change in best fitting OH interaction probability was dependent on model, becoming 0.12, 0.05 and 0.16 for opt2, opt4 and opt6 respectively. Overall fit quality was significantly reduced, with residual sums of squares 3-4 times higher than when the best-fitting threshold parameter was used.

Some text has been added on page 9 discussing these points.

All simulations were carried out in water, so all the free parameters (energy threshold, OH probability…) were determined for this material depending on the physic list used. In the literature, the importance of taking into account the composition DNA base pairs for this type of simulation has been highlighted. Indeed, for example, the paper of S. Zein et al (NIMB 2021) show the differences with water simulation and emphasize the importance to take into account the DNA base pairs composition to better estimate the radiobiological damages. I think this point should be mentioned and discussed on this manuscript.

We definitely agree with this point, since as the reviewer notes and is discussed towards the end of the discussion in the manuscript the exact distribution of energy deposits can be very significant for determining rates of simulated DNA damage, and these values can depend significantly on the material composition, with the possibility that some current parameters are ‘artifacts’ from the use of water as a material. Zein et al was referenced in support of this point in the original discussion but based on reviewer feedback this section has been revised and expanded to further emphasise this point.

See updated text on page 14.

You mentioned also the importance of the geometry used for simulations. It would be interesting to perform the same study with different geometries in order to quantify the impact of the geometry used on the number of damages.

This is quite a broad topic, as differences in dimensions, shapes, and physical properties assigned to each sub-volume can significantly change the predicted responses in the DNA. A comprehensive study of this is outside the scope of this manuscript, but collaborators have recently completed a manuscript on this topic which is also making its way through the review process, and should be available around the same time as this manuscript. A reference has been added to the final paragraph of the discussion at this point ([32], Bertolet et al) to the forthcoming article.

Reviewer 2 Report

The authors compared different Geant4-DNA physics models in predicting the SSB from I-125 with experimental data by Kandaiya. This paper was well written, and all the results are clearly shown in the figures. The analysis about the differences between model predictions and experimental data is sound. I would recommend the acceptance of this manuscript.

Can you please explain the difference between direct and indirect SSB?

Can you please briefly describe the experimental method in quantifying SSB by Kandaiya although they are not your work, because this will facilitate the readers to better understand your simulation results?

Author Response

We would like to thank the reviewer for their positive feedback on the manuscript, and their useful suggestions. We have revised the manuscript with respect to these suggestions, as described below. Changes in the updated manuscript are highlighted in red for clarity.

Can you please explain the difference between direct and indirect SSB?

As noted in the methods (section 2.3), direct SSBs are those which are caused by energy deposition directly within the DNA, while indirect breaks are those caused by interactions of radical species with the DNA. Text has been updated in the introduction to expand on this point (second paragraph, page 2), as well as other changes requested by reviewer 2 regarding the relationship between scavenging and direct/indirect damage (page 7 & 8)

Can you please briefly describe the experimental method in quantifying SSB by Kandaiya although they are not your work, because this will facilitate the readers to better understand your simulation results?

We’re happy to include more details here, and have expanded the text in section 2.1 of the methods (page 3) to include more precise detail on the steps used to quantify the yields of different DNA fragments. We’d be happy to include any further detail the reviewer thinks would be useful.

Round 2

Reviewer 1 Report

The authors have addressed all my concerns. I recommend this manuscript for publication.